# TF-Coder: Program Synthesis for Tensor Manipulations

Kensen Shi
Google Brain
United States
kshi@google.com

David Bieber
Google Brain
United States
dbieber@google.com

Rishabh Singh
Google Brain
United States
rising@google.com

## Abstract

Deep learning frameworks such as TensorFlow and PyTorch come with steep learning curves. We present a tool called TF-Coder for programming by example in TensorFlow. It uses a bottom-up weighted enumerative search with learned models that prioritize relevant operations. TF-Coder solves 63 of 70 real-world tasks within 5 minutes, often achieving superhuman performance—finding solutions that are simpler than those written by TensorFlow experts, in less time.

## 1 Introduction

The widespread success of deep learning is partially attributed to frameworks such as TensorFlow [Abadi et al. 2016] and PyTorch [Paszke et al. 2017] that help machine learning researchers develop models more effectively. However, these frameworks have a steep learning curve, since they offer a huge amount of functionality. Most deep learning models require various *tensor manipulations* for data cleaning or custom loss functions and metrics, but because there are about 500 tensor-manipulating operations in TensorFlow, finding the right ones to use for a given task can be a challenge itself.

We present TF-Coder, a programming by example system to automatically synthesize tensor manipulation programs from input/output examples and natural language descriptions. The synthesis algorithm builds upon the bottom-up enumerative algorithm proposed in TRANSIT [Udupa et al. 2013]. We introduce per-operation weights to the prior algorithm allowing TF-Coder to enumerate over expressions in order of increasing complexity, and a novel and efficient *operation filtering system* that enforces arbitrary preconditions imposed by TensorFlow operations. Finally, we combine predictions from multiple machine learning models to prioritize operations during the search, which helps tailor the search to fit the particular synthesis task at hand.

We evaluate TF-Coder on 70 real-world tensor transformation tasks from StackOverflow and from an industrial setting. TF-Coder can successfully synthesize solutions to 63 tasks in 17 seconds on average, while TRANSIT only solves 39

Authors' addresses: Kensen Shi, Google Brain, United States, kshi@google.com; David Bieber, Google Brain, United States, dbieber@google.com; Rishabh Singh, Google Brain, United States, rising@google.com.

2018. 2475-1421/2018/1-ART1 $15.00
https://doi.org/

tasks. Moreover, incorporating the trained models leads to significantly faster synthesis times (35.4% faster on average). We also observed that TF-Coder often produces solutions that are simpler and more elegant than those written by TensorFlow experts (including the authors of this paper).

## 2 Synthesis with Enumerative Search

We assume a given task specification $\phi = \{(\mathcal{I}, \mathcal{O}), D, C\}$, where $(\mathcal{I}, \mathcal{O})$ is an input/output example, i.e., a list of input tensors $\mathcal{I}$ and the corresponding output tensor $\mathcal{O}$, $D$ is an optional natural language description of the task, and $C$ is an optional set of constants that may be useful for the task. Our goal is to synthesize a program $P \in \mathcal{D}$ where $P(\mathcal{I}) = \mathcal{O}$. The domain of programs $\mathcal{D}$ considered by TF-Coder consists of single-line TensorFlow expressions, which may contain input variables, Python literals, TensorFlow function calls, and various Python operations such as indexing and slicing. In total, TF-Coder currently supports 134 different operations.

***Weighted Value Search***   TF-Coder's enumerative search is presented in Algorithm 1. The search expressions in order of increasing *weight*, which represents the expression's complexity. Operations and initial values (input tensors and constants) have associated weights, and an expression's weight is defined to be the sum of the weights of the operations and initial values used in that expression. We manually assigned weights for each of TF-Coder's supported operations, taking into consideration how common, useful, or complex it is. These weights allow TF-Coder to prioritize simple and useful operations in its search, which is crucial for enabling TF-Coder to handle so many different operations—niche operations are given higher weight so they can still be used if necessary, without causing much slowdown if they are not needed in a particular problem.

Starting with initial values, the algorithm generates expressions in order of increasing weight. For a given target weight, it enumerates over all supported operations and all ways of choosing arguments from previously-seen values such that the result has the desired weight. Every value produced in this way stores references to the operation and the arguments, so that any value can recursively reconstruct its code representation. As soon as TF-Coder encounters a value that is equal to the desired output tensor, it outputs the value's code representation as a solution.

---

**Algorithm 1**    TF-Coder's Synthesis Algorithm

---

**Input:** I/O example $(\mathcal{I}, O)$, NL description $D$, constants $C$
**Output:** A program $P$ such that $P(\mathcal{I}) = O$
**Auxiliary Data:** Supported operations $Ops$ (each $op$ has argument filters $\{op.f_i\}$ and combination filter $op.F$), and models $M_{io}$ and $M_{nl}$ conditioned on I/O examples and natural language, respectively

1:   $m_{io} \leftarrow M_{io}(\mathcal{I}, O)$          ▷ Model predictions
2:   $m_{nl} \leftarrow M_{nl}(D)$
3:   **for all** $op \in Ops$ **do**     ▷ Prioritize ops using models
4:      $op.weight \leftarrow \textsc{ReweightOp}(op, m_{io}, m_{nl})$
5:   $E \leftarrow \mathcal{I} \cup C$           ▷ Set of explored values
6:   $E \leftarrow E \cup \textsc{HeuristicConstants}(\mathcal{I}, O)$
7:   **for all** $v \in E$ **do**
8:      $v.weight \leftarrow \textsc{AssignWeightByOrigin}(v)$
9:   **for** $W = 1, 2, \ldots$ **do**     ▷ Weight of expressions
10:    **for all** $op \in Ops$ **do**
11:      $n \leftarrow op.arity$
12:      $w \leftarrow op.weight$
13:      **for all** $[w_1, \ldots, w_n]$      ▷ Argument weights
14:        s.t. $\sum_i w_i = W - w, \;\; w_i \in \mathbb{Z}^+$ **do**
15:       **for** $i = 1, \ldots, n$ **do**   ▷ Collect argument choices
16:         $A_i \leftarrow \{e \in E \mid e.weight = w_i \wedge op.f_i(e)\}$
17:       **for all** $[a_1, \ldots, a_n] \in \Pi_i A_i$ **do**
18:        **if** $\neg op.F([a_1, \ldots, a_n])$ **then**
19:         **continue**
20:        $V \leftarrow \textsc{Execute}(op, [a_1, \ldots, a_n])$
21:        **if** $V \notin E$ **then**       ▷ New value discovered
22:         $V.weight \leftarrow W$
23:         $V.history \leftarrow (op, [a_1, \ldots, a_n])$
24:         $E \leftarrow E \cup \{V\}$
25:        **if** $V = O$ **then**        ▷ Solution found
26:         **return** $\textsc{CodeExpression}(V)$

---

**Operation Filtering**    When the search enumerates argument lists for a particular operation, a full Cartesian product of argument choices may be very large, even though very few argument lists actually meet preconditions required by the operation. We introduce a flexible two-stage operation filtering approach to avoid enormous Cartesian products and reduce the number of errors thrown by operations.

The first stage occurs independently for each argument of the operation. An "argument filter" ($op.f_i$ in Algorithm 1) is a function that determines whether a value is an acceptable choice for a particular argument of an operation. For example, the `tf.argmax(input, axis)` operation requires that `input` is numeric tensor, so an argument filter for `input` would reject boolean tensors. Argument filters greatly reduce the size of the Cartesian product of argument choices.

The second stage of operation filtering checks constraints that involve multiple arguments. A "combination filter" ($op.F$ in Algorithm 1) for an operation is a function that determines

whether an argument list is suitable for the operation. For example, the combination filter for the `tf.argmax(input, axis)` operation would remove argument lists where `axis` is out-of-bounds for `input`. In this way, TF-Coder avoids executing expensive operations that would fail anyway.

For the difficult task in Figure 2c, the two-stage filtering strategy eliminates on average 98.6% of all potential candidate programs for a *single* operation, and a correct program must satisfy the constraints for *all* of its operations.

## 3   Learning to Guide the Search

Operation weights allow TF-Coder to prioritize simple and useful operations. Weights can also be modified to fit the specific synthesis problem, instead of having static weights.

TF-Coder uses two machine learning models to guide the search: a neural model conditioned on features of the example tensors, and a bag-of-words model conditioned on the natural language description. Each model's predictions are used to prioritize chosen operations by multiplying their weights by a constant (0.75 in our experiments), rounding to the nearest integer. Then, the enumerative search described in Section 2 is run using the modified operation weights.

***Tensor Features Model***    We train a neural model that learns a Bernoulli distribution over each operation, conditioned on features of the example tensors. Human experts can often recognize useful operations by observing patterns in the examples, e.g., if one tensor contains small nonnegative integers, they may represent indices into another tensor, especially if the output tensor contains entries that are found in the input tensors. Our goal is to learn such pattern-recognition capabilities with this model.

Due to the lack of a large supervised dataset containing real TensorFlow programs together with corresponding I/O examples, we train our model on a dataset generated synthetically by running our enumerative search algorithm on randomly-generated inputs, using values encountered during the search as outputs. We featurize example tensors and use feedforward layers to produce a logit for each operation.

We experiment with various forms of loss functions. One is standard sigmoid cross entropy loss averaged over the operations. However, as each example only uses a few operations, the dataset is overwhelmingly negative, leading to overly conservative predictions. Thus, we also implement a differentiable $F_\beta$ metric [van Rijsbergen 1979] as a loss function to achieve different balances in precision and recall. Because some operations appear in the synthetic dataset more often than others, we also try different weighting schemes, $w_i^{\max}$ or $w_i^{\text{mean}}$, so that whenever an operation is used, its loss term is scaled to the maximum or mean frequency of operations in the dataset. Between sigmoid cross entropy, $F_1$, and $F_2$ loss, combined with $w_i^{\max}$ or $w_i^{\text{mean}}$ weighting or no weighting, we have 9 different loss function variations.

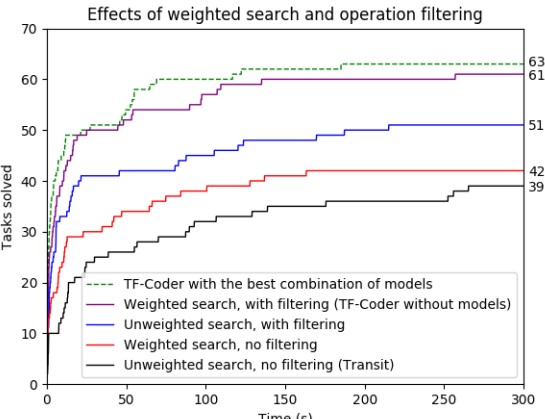

**Figure 1.** Ablation study for weighted search and operation filtering, showing the number of tasks that can be solved within some amount of time. Without these improvements, the algorithm reduces to that of the prior work TRANSIT.

**Natural Language Model**   We also train a model that prioritizes operations based on the natural language description of the task. We formulate this as a supervised multilabel classification problem: given a description, the model predicts whether each operation occurs in the solution.

Since we do not have a large dataset of TF-Coder queries paired with TensorFlow operations, we construct a training dataset from the TensorFlow documentation and from TensorFlow code on GitHub. From functions that use TensorFlow operations, we extract docstrings, comments, string literals, and variable names to use as a proxy for the task description, trying to predict which TensorFlow operations were actually used in the function.

We consider two classes of models: TF-IDF cosine similarity (the TF-IDF model), and naïve Bayes. Though the natural language in the constructed dataset often differs in structure from real TF-Coder task descriptions, we hypothesize that we can still learn from the *vocabulary* used in the dataset. So, we focus on these two bag-of-words models, rather than higher capacity models which would better fit the dataset but not generalize to the target domain of TF-Coder descriptions.

## 4   Experiments

We now present an evaluation of TF-Coder on a set of real-world benchmarks. We collected a benchmark set of 70 tensor manipulation tasks, including 50 drawn from StackOverflow questions and 20 real tasks encountered by TensorFlow users in an industrial setting. We aimed to include a diverse set of tasks as well as a wide range of task difficulties.

**Comparison to TRANSIT**   TF-Coder extends the search in TRANSIT [Udupa et al. 2013] in several important ways:

1. TF-Coder incorporates weights for operations and base values, while TRANSIT does not use weights.

2. TF-Coder uses a flexible operation filtering system that generalizes TRANSIT's type checking, which is insufficient for many TensorFlow operations.
3. TF-Coder uses two models to modify operation weights.

To evaluate the first two improvements, we run 4 variations of TF-Coder where we independently turn on or off weighting and operation filtering, without models.

The results of these 4 variations on our benchmarks are plotted in Figure 1. Both techniques in isolation lead to significant improvement over the TRANSIT algorithm, and their combination produces another large improvement. Overall, TF-Coder without any models can solve 61 of the 70 benchmark tasks within 5 minutes, while TRANSIT only solves 39 tasks. From the plot, it is clear that both techniques play a key role in TF-Coder's success.

**Perturbing Hardcoded Weights**   To see how TF-Coder is affected by our hardcoded operation weights, we ran TF-Coder where each operation's weight is multiplied by scaling factor between 0.8 and 1.25, drawn such that each operation is equally likely to have increased versus decreased weight. Compared to TF-Coder with the original weights, the run with perturbed weights solved 1 fewer task and was 6.2% slower on average. The perturbed run is only slightly worse, which implies that our weights are chosen reasonably well, but their exact values are not critical to TF-Coder's success.

**Effect of the Learned Models**   TF-Coder without learned models solves 61 tasks. We first experimented with adding a tensor features model and a natural language model in isolation, and then we ran TF-Coder with the combination of models. We use the metrics "total speedup" which compares the sum of the solve times for the 61 tasks solved without models, and "average speedup" which is the average over tasks of the per-task speedup. Total speedup is biased toward the performance on long-running tasks, while the average speedup is representative of all tasks (even easy tasks where the absolute time saved might not even be noticeable).

The best tensor features model uses the $F_1$ loss function with the $w_i^{\max}$ weighting scheme, solving 63 tasks with 31.3% total speedup and 24.9% average speedup. The best NL model uses TF-IDF, solving 61 tasks with 7.4% total speedup and 16.1% average speedup. The combination of models solves 63 tasks with 37.3% total speedup and 35.4% average speedup.

Even though TF-Coder without models is highly optimized and powerful, we observe significant speedups and extra solved tasks when using learned models to adjust the search to fit the current task. It is also promising that the model combinations perform significantly better than the individual models alone, showing that complementary models can jointly influence the search with compounding benefits.

**Comparison to StackOverflow**   We compare TF-Coder's performance with that of the StackOverflow community. We found that, among the 50 StackOverflow questions, 47 had

answers but only 32 were *correct* answers, with a median answer-posting time of 31 minutes. In comparison, TF-Coder is able to solve 44 of the StackOverflow tasks within 5 minutes, with a median solve time of 1.6 seconds.

***Comparison to Human Programmers*** We evaluated TF-Coder on 6 new benchmark tasks from StackOverflow, since our choices for the two models was influenced by their aggregate performance on the original set of 70 benchmarks.

We had volunteers with TensorFlow experience solve the problems manually (each problem attempted by 7 people), using whatever tools and resources they would normally use in such a scenario (except TF-Coder). We imposed a time limit of 5 minutes, not including the time spent reading the problem statement. We then asked 3 different people with some familiarity with TF-Coder to create an I/O example for each problem, without using TF-Coder. We declare an example to be "good" if it causes TF-Coder to produce a correct solution within 5 minutes.

Overall, the problems were difficult for the human volunteers to solve, with only 50% of solve attempts resulting in a correct solution, with a median solve time of 248 seconds. In comparison, 61% of examples given to TF-Coder were "good" with a median solve time of 23.3 seconds, and after a single revision, 94% of examples were "good." Even if the median example-creation time of 87 seconds were added to TF-Coder's solve times, TF-Coder would still be faster than the human times for 5 of the 6 problems. Thus, we conclude that TF-Coder achieves superhuman synthesis times on these new StackOverflow tasks.

***Comparison to AUTOPANDAS*** AUTOPANDAS [Bavishi et al. 2019] is a program synthesis tool for the Pandas library. This is an *informal* comparison due to differences in hardware, benchmark tasks, and TensorFlow versus Pandas.

Considering the 52 TF-Coder benchmarks that we could convert to Pandas format, the public AUTOPANDAS tool only solves 2 tasks within its 30 second timeout, but TF-Coder solves 36 tasks within 30 seconds. Next, considering the 16 AUTOPANDAS benchmarks that we could solve manually in TensorFlow, TF-Coder solves 14/16 in 20 minutes with a mean time of 18.3 seconds, while AUTOPANDAS solves 11/16 with a mean time of 36.6 seconds (according to their paper).

***Sample of Synthesized Programs*** Figure 2 shows examples of interesting problems that TF-Coder is able to solve.

TF-Coder often finds elegant solutions to problems using unconventional combinations of operations that the programmer might not have considered. For example, TF-Coder solves Problem (a) by composing `tf.math.bincount` with `tf.sequence_mask`, which is quite unconventional with zero instances in public GitHub repositories, StackOverflow posts, or Google search results.

Problem (b) was a real task encountered by an author of this paper (on a different project), and their initial solution

```
# Convert tensor into pairs for SparseTensor indexing.
in1 = [0, 0, 0, 1, 3, 3]
out = [[0, 0], [0, 1], [0, 2], [1, 0], [3, 0], [3, 1]]
# Solution found in 2.6 seconds
tf.cast(tf.where(tf.sequence_mask(
    tf.math.bincount(in1))), tf.int32)
```

**(a)** This task is solved using an unusual composition of operations.

```
# Reorder segments.
in1 = [10, 20, 30, 40, 50, 13, 17, 19, 21, 22, 23]
in2 = [1, 1, 1, 1, 1, 0, 0, 0, 2, 2, 2],
out = [13, 17, 19, 10, 20, 30, 40, 50, 21, 22, 23]
# Solution found in 2.5 seconds
tf.gather(in1, tf.argsort(in2, axis=0, stable=True))
```

**(b)** TF-Coder can help users learn about uncommon operations such as `tf.argsort`, which is crucial for this problem.

```
# Find the indices of all elements.
in1 = [32, 53, 45, 38, 29, 89, 64, 23]
in2 = [38, 53, 89, 38, 32, 64]
out = [3, 1, 5, 3, 0, 6]
# Solution found in 65.6 seconds
tf.cast(tf.argmax(tf.cast(tf.equal(in1, tf.expand_dims(
    in2, 1)), tf.int32), axis=1), tf.int32)
```

**(c)** This StackOverflow task requires a particularly long solution.

**Figure 2.** TF-Coder's results on selected tasks.

was a 12-line function using 18 TensorFlow operations. Incredibly, TF-Coder solves this problem in 2.5 seconds using only 2 operations, demonstrating that TF-Coder can find simpler solutions than expert humans.

Problem (c) shows one of the largest solutions found by TF-Coder, involving 5 TensorFlow operations and 11 total nodes in the expression tree. In TF-Coder's search space for this problem, there are $2.3 \times 10^{18}$ expressions of that size. Even so, TF-Coder finds this solution in only one minute.

## 5 Related Work

Prior program synthesis approaches can be broadly classified based on the underlying search mechanism: enumerative [Udupa et al. 2013], constraint-based [Solar-Lezama et al. 2006], and stochastic [Schkufza et al. 2013; Shi et al. 2018]. Applying constraint-based techniques to the TensorFlow domain would require a huge effort of modeling semantics of TensorFlow operations. TF-Coder builds on top of the bottom-up enumerative search from TRANSIT [Udupa et al. 2013], adding expression weights, operation filtering, and learned models to adjust weights based on the problem.

AUTOPANDAS [Bavishi et al. 2019] uses GNNs to synthesize Pandas programs by representing DataFrame examples as graphs with edges connecting equal cells, but this is not as effective for tensor manipulations, as many common mathematical operations would break the cell-equivalence edges.

AutoML systems search for ML model architectures that achieve high accuracy on a training dataset [Cambronero

and Rinard 2019; Zoph and Le 2016]. In contrast, TF-Coder instead uses examples to synthesize implementations of tensor manipulations that ML practitioners might perform.

## 6 Conclusion

In this paper, we presented TF-Coder, a synthesis tool for automatically generating tensor manipulation programs in TensorFlow from examples and natural language. We found that TF-Coder solves real-life tensor manipulation problems within seconds and outperforms human programmers.

For more details, please refer to our full paper [Shi et al. 2020].

## Acknowledgments

The authors thank Charles Sutton and the other members of the program synthesis team at Google Brain for helpful discussions.

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
