# OpenReview forum: "TF-Coder: Program Synthesis for Tensor Manipulations"
_NeurIPS.cc/2020/Workshop/CAP — NeurIPS 2020 CAP Workshop_

### Official Review · AnonReviewer1 · 2020-10-31
**Great, useful tool**

**Rating:** 7
**Confidence:** 5

**Review:**

The paper describes a tool for automatic synthesis on TensorFlow expressions from input-output examples of tensors and an optional NL description. The tool performs a bottom-up enumerative search guided by manually crafted weights and two guided models - one for embedding examples and another for embedding the NL description.

The TF-Coder tool is clearly useful, well designed, and fast. Its accuracy on the collected dataset of 70 benchmarks is impressive, especially with a median search time of just a few seconds. The individual components of the tool (bottom-up search, preconditions, search-guiding models) are well-known and not necessarily novel scientific contributions, but their application to synthesis of TensorFlow expressions is definitely an excellent work for this workshop.

From the perspective of scientific novelty and presentation, there are a couple obvious qualms with the approach.
First, the weights are selected heuristically. The authors have a limited evaluation of their impact and find that it's not _too_ brittle, but they are obviously impactful (as Figure 1 shows). One could imagine learning a weight distribution on the operators similarly to the way the search-guiding model is currently trained on synthetic data, DeepCoder-style. Or one could imagine starting with a language model of TF ops (trained from all the TF code in the wild) and then only using the search-guiding model to adjust these frequency weights. If some variant of this performs not too much worse than the expertly-crafted weights, it would open the door to generalizing the approach to other domains with little to no manual work.

The "operation filters" presented in great detail in the paper are nothing more than _preconditions_ on each op. The author's example of `argmax` can be accompanied by a precondition `φ(input, axis): input.dtype == tf.float32 and axis in range(tf.rank(input))`. This terminology is familiar to everyone in software engineering and would greatly simplify the paper.


Questions:
* Where do constants come from? Does the user have to specify them?
* The guiding models don't add much value over weight search. Do you have any intuition as to why?

---

### Decision · Program_Chairs · 2020-11-03

**Decision:**

Accept

**Comment:**

The PCs have decided to accept this paper.